# Towards Deep Conversational Recommendations

Raymond Li[1, 2], Samira Ebrahimi Kahou[1, 3], Hannes Schulz[3], Vincent Michalski[4, 5],
Laurent Charlin[5, 6], and Chris Pal[1, 2, 5]

[1]Ecole Polytechnique de Montréal    [2]Element AI    [3]Microsoft Research Montreal
[4]Université de Montréal    [5]Mila    [6]HEC Montréal

## Abstract

There has been growing interest in using neural networks and deep learning techniques to create dialogue systems. Conversational recommendation is an interesting setting for the scientific exploration of dialogue with natural language as the associated discourse involves goal-driven dialogue that often transforms naturally into more free-form chat. This paper provides two contributions. First, until now there has been no publicly available large-scale dataset consisting of real-world dialogues centered around recommendations. To address this issue and to facilitate our exploration here, we have collected REDIAL, a dataset consisting of over 10,000 conversations centered around the theme of providing movie recommendations. We make this data available to the community for further research. Second, we use this dataset to explore multiple facets of conversational recommendations. In particular we explore new neural architectures, mechanisms, and methods suitable for composing conversational recommendation systems. Our dataset allows us to systematically probe model sub-components addressing different parts of the overall problem domain ranging from: sentiment analysis and cold-start recommendation generation to detailed aspects of how natural language is used in this setting in the real world. We combine such sub-components into a full-blown dialogue system and examine its behavior.

## 1   Introduction

Deep-learning-based approaches to creating dialogue systems provide extremely flexible solutions for the fundamental algorithms underlying dialogue systems. In this paper we explore fundamental algorithmic elements of *conversational recommendation* systems through examining a suite of neural architectures for sub-problems of conversational recommendation making.

It is well known that deep learning techniques require considerable amounts of data to be effective. Addressing this need, we provide a new dataset of 10,000 dialogues to the community to facilitate the study of discourse with natural language when making recommendations is an explicit goal of the exchange. Our setting of interest and our new dataset, named REcommendations through DIALog (REDIAL)[1], are centered around conversations about movies where one party in the conversation is seeking recommendations and the other party is providing recommendations. Our decision for focusing on this domain is motivated in part by the following.

A good discussion with a friend, librarian, movie rental store clerk or movie fan can be an enjoyable experience, leading to new ideas for movies that one might like to watch. We shall refer to this general setting as *conversational movie recommendation*. While dialogue systems are sometimes characterized as falling into the categories of goal-directed dialogue vs chit-chat, we observe that discussions about movies often combine various elements of chit-chat, goal-directed dialogue, and even question answering in a natural way. As such the practical goal of creating conversational

recommendation systems provides an excellent setting for the scientific exploration of the continuum between these tasks.

This paper makes a number of contributions. First we provide the only real-world, two-party conversational corpus of this form (that we are aware of) to the community. We outline the data-collection procedure in Section 3. Second, we use this corpus to systematically propose and evaluate neural models for key sub-components of an overall conversational recommendation system. We focus our exploration on three key elements of such a system, consisting of: 1) Making recommendations; we examine sampling based methods for learning to make recommendations in the cold-start setting using an autoencoder [1]. We present this model in Section 4.3 and evaluate it in Section 5. Prior work with such models has not examined the cold-start setting which must be addressed in our dialogue set-up. 2) Classifying opinions or the sentiment of a dialogue participant with respect to a particular movie. For this task throughout the dialogue whenever a new movie is discussed we instantiate an RNN-based sentiment-prediction model. This model is used to populate the autoencoder-based recommendation engine above. We present this model component and our analysis of its behavior and performance in Sections 4.2 and 5 respectively. 3) We compose the components outlined above into a complete neural dialogue model for conversation and recommendation. For this aspect of the problem we examine a novel formulation of a hierarchical recurrent encoder-decoder (HRED) model [2] with a switching mechanism inspired from Gulcehre et al. [3] that allows suggested movies to be integrated into the model for the dialogue acts of the recommender. As our new dataset is relatively small for neural network techniques, our modular approach allows one to train sub-components on other larger data sources, whereas naïvely training end-to-end neural models from scratch using only our collected dialogue data can lead to overfitting.

## 2 Related Work

While we are aware of no large scale public dataset of human-to-human dialogue on the subject of movie recommendations, we review some of the most relevant work of which we are aware below. We also review a selection of prior work on related methods in Section 4 just prior to introducing each component of our model.

Dodge et al. [4] introduced four movie dialogue datasets comprising the Facebook Movie Dialog Data Set. There is a QA dataset, a recommendation dataset, and a QA + recommendation dataset. All three are synthetic datasets built from the classic MovieLens ratings dataset [5][2] and Open Movie Database[3]. Others have also explored procedures for generating synthetic dialogues from ratings data [6]. The fourth dataset is a Reddit dataset composed of around 1M dialogues from the movie subreddit[4]. The recommendation dataset is the closest to what we propose, however it is synthetically generated from natural language patterns, and the answers are always a single movie name. The Reddit dataset is also similar to ours in the sense that it consists of natural conversations on the topic of movies. However, the exchanges are more free-form and obtaining a good recommendation is not a goal of the discourse.

Krause et al. [7] introduce a dataset of *self dialogues* collected for the Amazon Alexa Prize competition[5], using Amazon Mechanical Turk (AMT). The workers are asked to imagine a conversation between two individuals on a given topic and to play both roles. The topics are mostly about movies, music, and sport. The conversations are not specifically about movie recommendations, but have the advantage of being quite natural, compared to the Facebook Movie Dialog Data Set. They use this data to develop a chat bot. The chat bot is made of several components, including: a rule-based component, a matching-score component that compares the context with similar conversations from the data to output a message from the data, and a (generative) recurrent neural network (RNN). They perform human evaluation of the matching-score component.

Some older work from the PhD thesis of Johansson [8] involved collecting a movie recommendation themed dialogue corpus with 24 dialogues, consisting of 2684 utterances and a mean of 112 utterances per dialogue. In contrast, our corpus has over 10k conversations and 160k utterances. See Serban et al. [9] for an updated survey of corpora for data-driven dialogue systems.

The recommender-systems literature has also proposed models for conversational systems. These approaches are goal-oriented and combine various different modules each designed (and trained) independently [10, 11]. Further, these approaches either rely on tracking the state of the dialogue

using slot-value pairs [12, 13] or focus on different objectives such as minimizing the number of user queries to obtain good recommendations [14]. Other approaches [15, 16, 17, 18] use reinforcement learning to train goal-oriented dialogue systems. Sun and Zhang [18] apply it to conversational recommendations: a simulated user allows to train the dialogue agent to extract the facet values needed to make an appropriate recommendation. In contrast, we propose a conditional generative model of (natural language) recommendation conversations and our contributed dataset allows one to both train sub-modules as well as explore *end-to-end* trainable models.

## 3  REDIAL dataset collection

Here we formalize the setup of a conversation involving recommendations for the purposes of data collection. To provide some additional structure to our data (and models) we define one person in the dialogue as the *recommendation seeker* and the other as the *recommender*. To obtain data in this form, we developed an interface and pairing mechanism mediated by Amazon Mechanical Turk (AMT). Our task setup is very similar to that used by Das et al. [19] to collect dialogue data around an image guessing game, except that we focus on movie recommendations. We pair up AMT workers and give each of them a role. The movie seeker has to explain what kind of movie he/she likes, and asks for movie suggestions. The recommender tries to understand the seeker's movie tastes, and recommends movies. All exchanges of information and recommendations are made using natural language.

We add additional instructions to improve the data quality and guide the workers to dialogue the way we expect them to. We ask to use formal language and that conversations contain roughly ten messages minimum. We also require that at least four different movies are mentioned in every conversation. Finally, we ask to converse only about movies, and notably not to mention Mechanical Turk or the task itself. See Figure 4 in the supplementary material for a screen-shot of the interface.

In addition, we ask that every movie mention is tagged using the '@' symbol. When workers type '@', the following characters are used to find matching movie names, and workers can choose a movie from that list. This allows us to detect exactly what movies are mentioned and when. We gathered entities from DBpedia that were of type `<http://dbpedia.org/ontology/Film>` to obtain a list of movies, but also allow workers to add movies to the list if it is not present already. We also obtained movie release dates from DBpedia. Note that the year or release date of a movie can be essential to differentiate movies with the same name, but released at different dates.

Workers are (separately from the on-going discussion) asked three questions for each movie: (1) Whether the movie was mentioned by the seeker, or was a suggestion from the recommender ("suggested" label); (2) Whether the seeker has seen the movie ("seen" label): one of *Seen it*, *Haven't seen it*, or *Didn't say*; (3) Whether the seeker liked the movie or the suggestion ("liked" label): one of *Liked*, *Didn't like*, *Didn't say*. We will refer to these additional labels as *movie dialogue forms*. Both workers have to answer these forms even though it really concerns the seeker's movie tastes. We use those ratings to validate data collection, the two workers agreeing in the forms being generally an indicator for conscientious workers. Ideally, the two participants would give the same answer to every form, but it is possible that their answers do not coincide (because of carelessness, or dialogue ambiguity). The dataset released provides both workers' answers. The movie dialogue forms therefore allow us to evaluate sub-components of an overall neural dialogue system more systematically, for example one can train and evaluate a sentiment analysis model directly using these labels. We believe that predicting sentiment from dialogues poses an interesting sub-challenge within conversational recommendation, as the sentiment can be expressed in a question-answer form over several dialogue utterances.

In each conversation, the number of movies mentioned varies, so we have different numbers of movie dialogue form answers for each conversation. The distribution of the different classes of the movie dialogue form is shown in Table 1a. The liked/disliked/did not say label is highly imbalanced. This is standard for recommendation data [20], since people are naturally more likely to talk about movies that they like, and the recommender's objective is to recommend movies that the seeker is likely to like. Table 1b shows an example of conversation from the dataset.

For the AMT HIT we collect data in English and restrict the data collection to countries where English is the main language. The fact that we pair workers together slows down the data collection since two people must be online at the same time to do the task, so a good amount of workers is required to make the collection possible. Meanwhile, the task is quite demanding, and we have to select qualified

| | | | |
|---|---:|---|---|
| # conversations | 10006 | **HUMAN:** | hello, can i help you find a movie ? |
| # utterances | 182150 | **HRED:** | hi what kind of movies do you like |
| # users | 956 | **OURS:** | hi, how can i help you ? |
| # movie mentions | 51699 | **SEEKER:** | hello ! i would like to find a suspenseful, but clean family friendly movie . |
| seeker mentioned | 16278 | ... | ... |
| recommender suggested | 35421 | **SEEKER:** | 2001 : a space odyssey might be a great option. anything else that you would suggest ? |
| not seen | 16516 | **HUMAN:** | you can try planet of the apes the older one is quite suspenseful and family friendly . |
| seen | 31694 | **HRED:** | i haven't seen that one yet but i heard it was good. i haven't seen that one. have you seen the last house on the left ? |
| did not say | 3489 | | |
| disliked (4.9%) | 2556 | **OURS:** | star wars : the force awakens is also a good one return of the jedi all good movies |
| liked (81%) | 41998 | | |
| did not say (14%) | 7145 | | |

| | |
|---|---|
| **SEEKER:** | those sound good ! i 'm going to look into those movies. |
| **HUMAN:** | i hope you enjoy, have a nice one |
| **HRED:** | have you seen foxcatcher ? it 's about a man who has a rich guy. |
| **OURS:** | i hope i was able to help you find a good movie to watch |
| **SEEKER:** | thank you for your help ! have a great night ! good bye |

**Table 1a.** (Above) REDIAL data statistics. For the movie dialogue forms, the numbers shown represent the seeker's answers.

**Table 1b.** (Right) Conversation excerpts (HUMAN followed by response by SEEKER) and model outputs (OUR proposed approach compared to HRED a generic dialogue model [2]).

Note: We provide additional conversation examples and model outputs in the supplementary material.

workers. HIT reward and qualification requirement were decisive to get good conversation quality while still ensuring that people could get paired together. We launched preliminary HITs to find a compromise and finally set the reward to $0.50 per person for each completed conversation (so each conversation costs us $1, plus taxes), and ask that workers meet the following requirements: (1) Approval percentage greater than 95; (2) Number of approved HITs greater than 1000; and (3) They must be in the United States, Canada, the United Kingdom, Australia or New Zealand.

## 4 Our Approach

We aim at developing an agent capable of chatting with a partner and asking questions about their movie tastes in order to make movie recommendations. One might therefore characterize our system as a recommendation "chat-bot". The complete architecture of our approach is illustrated in Figure 1. Starting from the bottom of Figure 1, there are four sub-components: (1) A hierarchical recurrent encoder following the HRED [2] architecture, using general purpose representations based on the Gensen model [21]; (2) A switching decoder inspired by Gulcehre et al. [3], modeling the dialogue acts generated by the recommender; (3) After each dialogue act our model detects if a movie entity has been discussed (with the @identifier convention) and we instantiate an RNN focused on classifying the seeker's sentiment or opinion regarding that entity. As such there are as many of these RNNs as there are movie entities discussed in the discourse. The sentiment analysis RNNs are used to indicate the user opinions forming the input to (4), an autoencoder-based recommendation module [1]. The autoencoder recommender's output is used by the decoder through a switching mechanism. Some of these components can be pre-trained on external data, thus compensating for the small data size. Notably, the switching mechanism allows one to include the recommendation engine, which we trained using the significantly larger MovieLens data. We provide more details for each of these components below and describe the training procedure in the supplementary materials.

### 4.1 Our Hierarchical Recurrent Encoder

Our dialogue model is reminiscent of the hierarchical recurrent encoder-decoder (HRED) architecture proposed and developed in Sordoni et al. [2] and Serban et al. [22]. We reuse their hierarchical

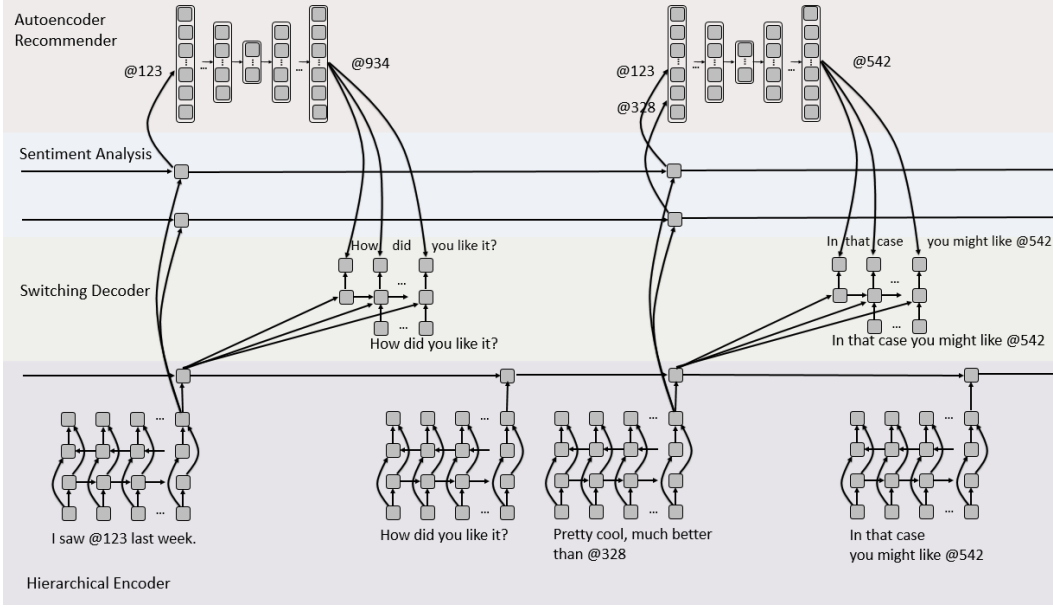

Figure 1: Our proposed model for conversational recommendations.

architecture, but we modify the decoder so that it can take explicit movie recommendations into account and we modify the encoder to take general purpose sentence (GenSen) representations arising from a bidirectional Gated Recurrent Unit (GRU) [23] as input. Since our new dataset here consists of about 10k dialogues (which is relatively small for deep learning techniques), we use pre-trained GenSen representations obtained from the encoder outlined in Subramanian et al. [21]. These representations have led to higher performance across a variety of new tasks in lower data regimes (e.g. with only 10k examples). We use the embeddings and first layer of the GenSen sentence encoder which are pre-trained on multiple language tasks and we keep them frozen during training of our model. To deal with the issue of how to process movies discussed in the dialogue using the @movie for movie entities, @movie tokens in the input are replaced by the corresponding word tokens for the title of the movie.

More formally, we model each utterance $U_m$ as a sequence of $N_m$ words $U_m = (w_{m,1}, ..., w_{m,N_m})$ where the tokens $w_{m,n}$ are either words from a vocabulary $V$ or movie names from a set of movies $V'$. We also use a scalar $s_m \in \{-1, 1\}$ appended to each utterance to indicate the role (recommender or seeker) such that a dialogue of $M$ utterances can be represented as $D = ((U_1, s_1), ..., (U_M, s_M))$. We use a GRU to encode utterances and dialogues. Given an input sequence $(\mathbf{i}_1, ..., \mathbf{i}_T)$, the network computes reset gates $\mathbf{r}_t$, input gates $\mathbf{z}_t$, new gates $\mathbf{n}_t$ and forward hidden state $\overrightarrow{\mathbf{h}}_t$ as follows:

$$\mathbf{r}_t = \sigma\left(\mathbf{W}_{ir}\mathbf{i}_t + \mathbf{W}_{hr}\overrightarrow{\mathbf{h}}_{t-1} + \mathbf{b}_r\right), \quad \mathbf{z}_t = \sigma\left(\mathbf{W}_{iz}\mathbf{i}_t + \mathbf{W}_{hz}\overrightarrow{\mathbf{h}}_{t-1} + \mathbf{b}_z\right)$$

$$\mathbf{n}_t = \tanh\left(\mathbf{W}_{in}\mathbf{i}_t + \mathbf{b}_{in} + \mathbf{r}_t \circ (\mathbf{W}_{hn}\overrightarrow{\mathbf{h}}_{t-1} + \mathbf{b}_{hn})\right), \quad \overrightarrow{\mathbf{h}}_t = (1 - \mathbf{z}_t) \circ \mathbf{n}_t + \mathbf{z}_t \circ \overrightarrow{\mathbf{h}}_{t-1}$$

Where the $\mathbf{W}_{**}$ and $\mathbf{b}_*$ are the learned parameters. In the case of a bi-directional GRU, the backward hidden state $\overleftarrow{\mathbf{h}}_t$ is computed the same way, but takes the inputs in a reverse order. In a multi-layer GRU, the hidden states of the first layer $\left(\overrightarrow{\mathbf{h}}_1^{(1)}, ..., \overrightarrow{\mathbf{h}}_T^{(1)}\right)$ (or the concatenation of the forward and backward hidden states of the first layer $\begin{bmatrix}\overrightarrow{\mathbf{h}}_1^{(1)} \\ \overleftarrow{\mathbf{h}}_1^{(1)}\end{bmatrix}, ..., \begin{bmatrix}\overrightarrow{\mathbf{h}}_T^{(1)} \\ \overleftarrow{\mathbf{h}}_T^{(1)}\end{bmatrix}$ for a bi-directional GRU) are passed as inputs to the second layer, and so on. For the utterance encoder words are embedded in a 2048 dimensional space. Each utterance is then passed to the sentence encoder bi-directional GRU. The final hidden state of the last layer is used as utterance representation $\mathbf{u} = \begin{bmatrix}\overrightarrow{\mathbf{h}}_T^{(-1)} \\ \overleftarrow{\mathbf{h}}_T^{(-1)}\end{bmatrix}$. We obtain a sequence of utterance representations $\mathbf{u}_1, ..., \mathbf{u}_M$. To assist the conversation encoder we append a

binary-valued scalar $s_m$ to each utterance representation $\mathbf{u}_m$, indicating if the sender is the seeker or the recommender. The sequence $\mathbf{u}'_1, ..., \mathbf{u}'_M$ is passed to the conversation encoder unidirectional GRU, which produces conversation representations at each step of the dialogue: $\mathbf{h}_1, ..., \mathbf{h}_M$.

## 4.2 Dynamically Instantiated RNNs for Movie Sentiment Analysis

In a test setting, users would not provide explicit ratings about movies mentioned in the conversation. Their sentiment can however be inferred from the utterances themselves. Therefore, to drive our autoencoder-based recommendation module we build a model that takes as input both the dialogue and a movie name, and predicts for that movie the answers to the associated movie dialogue form. We remind the reader that both workers answer the movie dialogue form, but it only concerns the seeker's movie tastes. It often happens that the two workers do not agree on all the answers to the forms. It may either come from a real ambiguity in the dialogue, or from worker carelessness (data noise). So the model predicts different answers for the seeker and for the recommender. For each participant it learns to predict three labels: the "suggested" label (binary), the "seen" label (categorical with three classes), the "liked" label (categorical with three classes) for a total of 14 dimensions.

Let us denote $\mathcal{D} = \{(x_i, y_i), i = 1..N\}$ the training set, where $x_i = (D_i, m_i)$ is the pair of a dialogue $D_i$ and a movie name $m_i$ that is mentioned in $D_i$ and

$$y_i = (\underbrace{y_i^{\text{sugg}}, y_i^{\text{seen}}, y_i^{\text{liked}}}_{\text{seeker's answers}}, \underbrace{y_i'^{\text{sugg}}, y_i'^{\text{seen}}, y_i'^{\text{liked}}}_{\text{recommender's answers}}), \tag{1}$$

are the labels in the movie dialogue form corresponding to movie $m_i$ in dialogue $D_i$. So if 5 movies were mentioned in dialogue $D$, this dialogue appears 5 times in a training epoch.

The model is based on a hierarchical encoder (Section 4.1). For sentiment analysis, we modify the utterance encoder to take the movie $m$ into account. After the first layer of the utterance encoder GRU (which is pre-trained), we add a dimension to the hidden states that indicate for each word if it is part of a movie mention. For example if we condition on the movie *The Sixth Sense*, then the input ["<s>", "you", "would", "like", "the", "sixth", "sense", ".", "</s>"] produces the movie mention feature: [0, 0, 0, 0, 1, 1, 1, 0, 0]. The utterance and conversation encoding continue as described in Section 4.1 afterwards, producing dialogue representations $\mathbf{h}_1, ..., \mathbf{h}_M$ at each dialogue step.

The dialogue representation at the last utterance $\mathbf{h}_M$ is passed in a fully connected layer. The resulting vector has 14 dimensions. We apply a sigmoid to the first component to obtain the predicted probability that the seeker answered that the movie was suggested by the recommender $o_i^{\text{sugg}}$. We apply a softmax to the next three components to obtain the predicted probabilities for the seeker's answer in the not-seen/seen/did-not-say variable $o_i^{\text{seen}}$. We apply a softmax to the next three components to obtain the predicted probabilities for the seeker's answer in the disliked/liked/did-not-say variable $o_i^{\text{liked}}$. The last 7 components are treated the same way to obtain the probabilities of answers according to the recommender $o'^{\text{sugg}}, o'^{\text{seen}}, o'^{\text{liked}}$. We denote the parameters of the neural network by $\theta$ and $o_i = f_\theta(x_i) = \left(o_i^{\text{sugg}}, o_i^{\text{seen}}, o_i^{\text{liked}}, o_i'^{\text{sugg}}, o_i'^{\text{seen}}, o_i'^{\text{liked}}\right)$, the prediction of the model. We minimize the sum of the three corresponding cross-entropy losses.

## 4.3 The Autoencoder Recommender

At the start of each conversation, the recommender has no prior information on the movie seeker (cold start). During the conversation, the recommender gathers information about the movie seeker and (implicitly) builds a profile of the seeker's movie preferences. Sedhain et al. [1] developed a user-based autoencoder for collaborative filtering (U-Autorec), a model capable of predicting ratings for users not seen in the training set. We use a similar model and pre-train it with MovieLens data [5].

We have $M$ users, $|V'|$ movies and a partially observed user-movie rating matrix $\mathbf{R} \in \mathbb{R}^{M \times |V'|}$. Each user $u \in \{1, ..., M\}$ can be represented by a partially observed vector $\mathbf{r}^{(u)} = \left(\mathbf{R}_{u,1}, ..., \mathbf{R}_{u,|V'|}\right)$. Sedhain et al. [1] project $\mathbf{r}^{(u)}$ in a smaller space with a fully connected layer, then retrieve the full ratings vector $\hat{\mathbf{r}}^{(u)} = h(\mathbf{r}^{(u)}; \theta)$ with another fully connected layer. So during training they minimize the following loss:

$$L_{\mathbf{R}}(\theta) = \sum_{u=1}^{M} \|\mathbf{r}^{(u)} - h(\mathbf{r}^{(u)}; \theta)\|_{\mathcal{O}}^2 + \lambda \|\theta\|^2 \tag{2}$$

where $\| \cdot \|_{\mathcal{O}}$ is the $L_2$ norm when considering the contribution of observed ratings only and $\lambda$ controls the regularization strength.

To improve the performance of this model in the early stage of performing recommendations (i.e. in cold-start setting) we train this model as a denoising auto-encoder [24]. We denote by $N_u$ the number of observed ratings in the user vector $\mathbf{r}^{(u)}$. During training, we sample the number of inputs kept $p$ uniformly at random in $\{1, ..., N_u - 1\}$. Then we draw $p$ inputs uniformly without replacement among all the observed inputs in $\mathbf{r}^{(u)}$, which gives us a noisy user vector $\tilde{\mathbf{r}}^{(u)}$. The term inside the sum of Equation 2 becomes $\|\mathbf{r}^{(u)} - h(\tilde{\mathbf{r}}^{(u)}; \theta)\|_{\mathcal{O}}^2$. The validation procedure is not changed: the complete input from the training set is used at validation or test time.

### 4.4 Our Decoder with a Movie Recommendation Switching Mechanism

Let us place ourselves at step $m$ in dialogue $D$. The sentiment analysis RNNs presented above predict for each movie mentioned so far whether the seeker liked it or not using the previous utterances. These predictions are used to create an input $\mathbf{r}_{m-1} \in \mathbb{R}^{|V'|}$ for the recommendation system. The recommendation system uses this input to produce a full vector of ratings $\hat{\mathbf{r}}_{m-1} \in \mathbb{R}^{|V'|}$. The hierarchical encoder (Section 4.1) produces the current context $\mathbf{h}_{m-1}$ using previous utterances. The recommendation vector $\hat{\mathbf{r}}_{m-1}$ and the context $\mathbf{h}_{m-1}$ are used by the decoder to predict the next utterance by the recommender.

For the decoder, a GRU decodes the context to predict the next utterance step by step. To select between the two types of tokens (words or movie names), we use a switch, as Gulcehre et al. [3] did for the *pointer softmax*. The decoder GRU's hidden state is initialized with the context $\mathbf{h}_{m-1}$, and decodes the sentence as follows: $\mathbf{h}'_{m,0} = \mathbf{h}_{m-1}$, $\mathbf{h}'_{m,n} = \text{GRU}(\mathbf{h}'_{m,n-1}, w_{m,n})$, $\mathbf{v}_{m,n} = \text{softmax}\left(\mathbf{W}\mathbf{h}'_{m,n}\right)$, $\mathbf{v}_{m,n} \in \mathbb{R}^{|V|}$ is the predicted probability distribution for the next token $w_{m,n+1}$, knowing that this token is a word. The recommendation vector $\hat{\mathbf{r}}_{m-1}$ is used to obtain a predicted probability distribution vector $\mathbf{v}'_{m,n} \in \mathbb{R}^{|V'|}$ for the next token $w_{m,n+1}$, knowing that this token is a movie name: $\mathbf{v}'_{m,n} = \text{softmax}(\hat{\mathbf{r}}_{m-1}) = \mathbf{v}'_{m,0} \quad \forall n$. Where we note that we use the same movie distribution $\mathbf{v}'_{m,0}$ during the whole utterance decoding. Indeed, while the recommender's message is being decoded, it does not gather additional information about the seeker's movie preferences, so the movie distribution should not change. A switching network conditioned on the context $\mathbf{h}_{m-1}$ and the hidden state $\mathbf{h}'_{m,n}$ predicts the probability $d_{m,n}$ that the next token $w_{m,n+1}$ is a word and not a movie name.

Such a switching mechanism allows to include an explicit recommendation system in the dialogue agent. One issue of this method is that the recommendations are conditioned on the movies mentioned in the dialogue, but not directly on the language. For example our system would be unable to provide recommendations to someone who just asks for "a good sci-fi movie". Initial experiments conditioning the recommendation system on the dialogue hidden state led to overfitting. This could be an interesting direction for future work. Another issue is that it relies on the use of the '@' symbol to mention movies, which could be addressed by adding an entity recognition module.

## 5 Experiments

We propose to evaluate the recommendation and sentiment-analysis modules separately using established metrics. We believe that these individual metrics will improve when modules are more tightly coupled in the recommendation system and thus provide a proxy to overall dialogue quality. We also perform an utterance-level human evaluation to compare responses generated by different models in similar settings.

Evaluating models in a fully interactive setting, conversing with a human, is the ultimate testing environment. However, evaluating even one response utterance at a time is an open challenge (e.g., [25]). We leave such evaluation for future work.

**Movie sentiment analysis performance:** We use the movie dialogue forms from our data to train and evaluate our proposed RNN-based movie sentiment analysis formulation. The results obtained for the seeker's answers and the recommender's answers are highly similar, thus we present the results only for the seeker's answers. We focus on understanding if models are able to correctly infer the

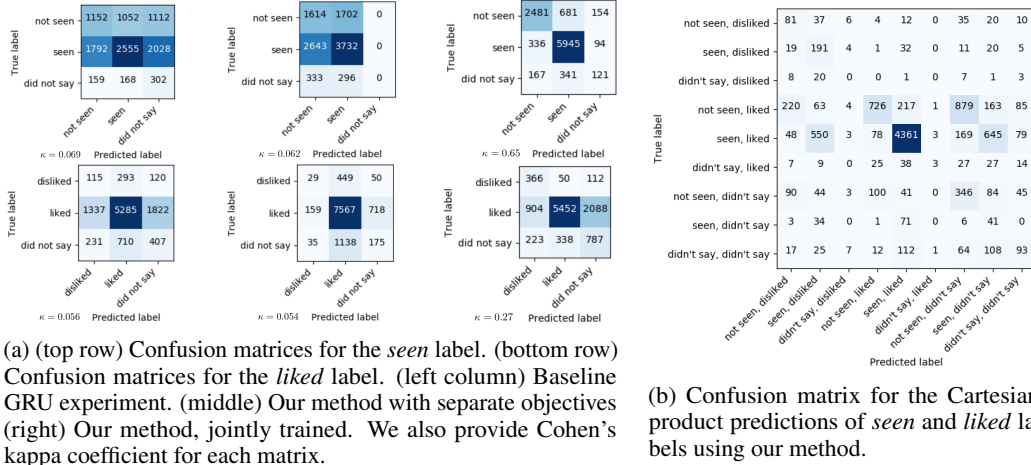

(a) (top row) Confusion matrices for the *seen* label. (bottom row) Confusion matrices for the *liked* label. (left column) Baseline GRU experiment. (middle) Our method with separate objectives (right) Our method, jointly trained. We also provide Cohen's kappa coefficient for each matrix.

(b) Confusion matrix for the Cartesian product predictions of *seen* and *liked* labels using our method.

Figure 2: Confusion matrices for movie sentiment analysis on the validation set.

*seen* vs *not seen*, and *liked* vs *not liked* assessments from the forms. Because of the class imbalance (i.e. $81\%$ of movies were liked, vs $4.9\%$ which were disliked), we weight the loss to compensate.

We compare with two simpler approaches. First, a baseline approach in which we pass the GenSen encodings of the sentences between the first and the last mention of a movie into a GRU layer. This is followed by a fully connected layer from the last hidden state. The prediction is made from the mean probability over all the sentences. Second, instead of using a single hierarchical encoder that is jointly trained to predict the three labels (suggested, seen and liked), we train the same model with only one of the three objectives (seen or liked) and demonstrate that the joint training regularizes the model. Figure 2a shows the confusion matrices for the *seen* and *liked* prediction tasks for, from left to right, the baseline model, our model trained on single objectives, and our method outlined in Section 4.2 and illustrated in the blue region of Figure 1. We also provide Cohen's kappa coefficient [26] for each model and prediction task. Cohen's kappa measures the agreement between the true label and the predictions. For each prediction task, our jointly trained model has a higher kappa coefficient than the two other baselies. The full confusion matrix for the Cartesian product of predictions is shown in Figure 2b. All results are on the validation set.

Table 2: RMSE for movie recommendations. RMSE is shown for ratings on a 0–1 scale. For the MovieLens experiment, we show the RMSE on a 0.5-5 scale in parenthesis.

| | | Experiments on REDIAL (validation RMSE) | |
|---|---|---|---|
| Training procedure | Experiments on MovieLens | No pre-training | Pre-trained on MovieLens |
| Standard Baseline | $0.181 \pm 0.001$ (0.813) | 0.125 | 0.075 |
| Denoising Autorec | $\mathbf{0.171} \pm 0.0006$ (0.769) | 0.127 | **0.072** |

**Movie recommendation quality:** We use the "latest" MovieLens dataset[6], that contains 26 million ratings across 46,000 movies, given by 270,000 users. It contains 2.6 times more ratings, but also across 4.6 times more movies than MovieLens-10 M, the dataset used in Sedhain et al. [1]. First, we evaluate the model on the MovieLens dataset. Randomly chosen user-item ratings are held out for validation and test, and only training ratings are used as inputs. Following Sedhain et al. [1], we sampled the training, validation, and test set in a 80-10-10 proportion, and repeated this splitting procedure five times, reporting the average RMSE.

We also examine how the model performs on the ratings from our data (REDIAL), with and without pre-training on MovieLens. This experiment ignores the conversational aspect of our data and focuses only on the like/dislike ratings provided by users. We chose to consider only the ratings given by the movie seeker, and to ignore the responses where he answered "did not say either way". We end up with a set of binary ratings for each conversation. To place ourselves in the setting of a recommender that meets a new movie seeker (cold-start setting), we consider each conversation as a

separate user. Randomly chosen conversations are held out for validation, and each rating, in turn, is predicted using all other ratings (from the same conversation) as inputs. We binarize the Movielens observations—they range between $0.5$ and $5$— for pre-training, by choosing a threshold that gives a similar distribution of $0$s and $1$s as in our data. Knowing that our data has $94.3\%$ of "liked" ratings, we chose a rating threshold of $2$: ratings higher or equal are considered as "liked", ratings lower are considered as "disliked". The binarized MovieLens dataset now has $93.7\%$ of "liked" ratings. In each experiment, for the two training procedures (standard and denoising), we perform a hyper-parameter search on the validation set.

Table 2 shows the RMSE obtained on the test set. In the experiment on the MovieLens dataset, the denoising training procedure brings a slight improvement on the standard training procedure. After pre-training on MovieLens, the performances of the models on our data is significantly improved.

**Overall dialogue quality assessment:** We run a user study to assess the overall quality of the responses of our model compared to HRED. Ten participants were each presented with ten complete real dialogues from our validation set, performing 56 ranking tasks– 1 for each recommender's utterance in those ten dialogues. At the point where the human recommender provided their response in the real dialogue we show: the text generated by our HRED baseline, our model, and the true response in a random order. The participant is asked to give the dialogue responses a rank from 1–3, with 1 being the best and 3 the worst. We allow ties so that multiple responses could be given the same rank (e.g., rankings of the form 1, 2, 2 were possible if the one response was clearly the best, but the other two were of equivalent quality). In Figure 3, we show the percentage of times that each model was given each ranking. The true response was ranked first 349 times, our model 267 times, and HRED 223 times.

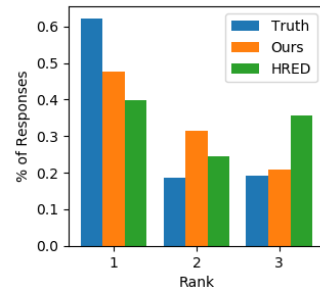

Figure 3: Results of human assessment of dialogue quality. The percentages are relative to the total number of ranking tasks, so that bars of the same color sum to 1.

# 6 Discussion and Conclusions

We presented REDIAL a new, high-utility dataset of real-world, human generated conversations around the theme of providing movie recommendations. 10,000 conversations will likely be insufficient to train an end-to-end neural model from scratch, we believe that this shortage of data is a systematic problem in goal-oriented dialogue settings and needs to be adressed at the modeling side. We use this dataset to explore a novel modular formulation of a fully neural architecture for conversational movie recommendations. The dataset has been collected in such a way that subtasks such as sentiment analysis and movie recommendation can be explored and evaluated separately or within the context of a complete dialogue system.

We introduced a novel overall architecture for this problem domain which leverages general purpose sentence representations and hierarchical encoder-decoder architectures, extending them with dynamically instantiated RNN models that drive an autoencoder-based recommendation engine. We find tremendous benefit from this modularization in that it allows one to pre-train the recommendation engine on other larger data sources specialized for the recommendation task alone. Further, our proposed switching mechanism allows one to integrate recommendations within a recurrent decoder, mixing high quality suggestions into the overall dialogue framework.

Our proposed architecture is not specific to movies and applies to other types of products, given that a conversational recommendation dataset is available in that domain. Our utterance-level evaluation compares the responses generated by different models in a given context, controlling for confounding variables to some extent. In that context, our model outperforms the HRED baseline. However, we did not yet evaluate whole conversations between our model and a human user. Future works could improve this evaluation setting by asking more precise questions to the human evaluators. Instead of asking which response is the best in a general way, we could ask for example which response provides the best recommendation given the context, or which is the most fluent. This would allow us to gain insight on what parts of the model could be improved.

## Footnotes

[1] `https://redialdata.github.io/website/`

[2] `https://grouplens.org/datasets/movielens/`     [3] `http://en.omdb.org`

[4] `http://reddit.com/r/movie`   [5] `https://developer.amazon.com/alexaprize`

[6] `https://grouplens.org/datasets/movielens/latest/`, retrieved September 2017.

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
