[Supplementary Material]

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

# A   Data collection interface

(a) Seeker interface

(b) Recommender interface

Figure 4: Data collection interface.

# B   Training Procedure

Two parts of our model are frozen after pre-training: the first layer of our utterance encoder (which uses the GenSen representations [21]) and the sentiment analysis module (after being pre-trained to predict the movie ratings). The recommender system is pre-trained on the Movielens dataset but will be fine-tuned with the rest of the network. Our model is then trained in teacher forcing using our dialogue dataset. The training objective is to maximize the likelihood of the next recommender utterance (which may contain words and movie entities). Thus, the switching decoder and the hierarchical encoder (except for the utterance encoder's first layer) are trained from scratch using our dataset. We retain 20% of the training conversations for the validation set. We optimize the

parameters with Adam [27] and a leaning rate of 0.001. At test time, we use a beam search of size 10 to generate the response.

## C Conversation samples

We show in tables 3, 4, 5, 6, 7, some randomly picked conversations from the validation set, and the responses generated by the HRED and by our model. At each utterance, we condition on the ground truth previous turns (labeled "SEEKER" and "HUMAN" in the tables) to generate the response.

Table 3: Sample conversation from validation set.

| | |
|---|---|
| SEEKER: | hi ! i 'm looking for an action filled movie similar to jurassic park |
| HUMAN: | ok , have you seen jurassic world ? |
| HRED: | hi ! i love that one ! have you seen the conjuring or the conjuring 2 ? |
| OURS: | hi what kind of movies do you like ? |
| SEEKER: | yes i loved that movie as well . are there any good movies without dinosaurs ? |
| HUMAN: | have you seen king kong ? |
| HRED: | no , i have n't seen it yet . have you seen the avengers ? it 's about a couple of friends who are their own . |
| OURS: | i have n't seen that one . i will have to check it out . have you seen jurassic park |
| SEEKER: | yes ! that was another great giant monster movie : ) |
| HUMAN: | what about jaws & amp ; jaws 2 ? oldies but goodies scary action ? suspense gives me the rumblies |
| HRED: | i have n't seen that one yet but i heard it was good . i have n't seen it yet . i have n't seen it yet . i have n't seen it yet . |
| OURS: | i have n't seen that one yet . i will have to check it out . what about walking with dinosaurs ? |
| SEEKER: | i absolutely loved the first one . one of my favorite movies ever : ) honestly i ca n't say i remember much about the second one . jaws 3-d was quite funny and jaws : the revenge was just weird . i do love suspense too ... |
| HUMAN: | i like suspense but sometimes i can barely handle it ! it gives me anxiety ! lol |
| HRED: | no , i have n't seen it yet . i will have to check it out . thanks for the chat . |
| OURS: | i have n't seen that one . i will have to check it out . i hope i helped . |
| SEEKER: | that 's understandable . as long as it 's not too gory i do n't generally have a problem with those things . |
| HUMAN: | well , it was great chatting with you ! have a great one ! |
| HRED: | i 'm glad i was able to help you today |
| OURS: | have you seen immortals ? it 's a war movie but it 's pretty good |
| SEEKER: | you too ! thanks ! |

Table 4: Sample conversation from validation set

| | |
|---|---|
| HUMAN: | hello . |
| HRED: | hi what kind of movies do you like |
| OURS: | hi , how can i help you ? |

| | |
|---|---|
| SEEKER: | hi |
| HUMAN: | what type of movie can i suggest to you ? |
| HRED: | what kind of movies do you like ? i 'm good |
| OURS: | what kind of movies do you like ? what kind of movies do you like ? |

| | |
|---|---|
| SEEKER: | i am looking for a slap stick comedy from the 80 's or 90 's |
| HUMAN: | not familiar with this , please give me a minute . |
| HRED: | have you seen the conjuring or the conjuring 2 ? it was really good and very funny |
| OURS: | what kind of movies do you like ? i 'm good , thanks for asking ! have a great day ! |

| | |
|---|---|
| SEEKER: | okay movies like airplane ! and airplane ii : the sequel |
| HUMAN: | oh , okay . i saw those and thought they were great . did you see caddyshack or dumb and dumber ? |
| HRED: | i have n't seen that one yet but i heard it was good . have you seen the last house on the left ? it 's pretty good . |
| OURS: | have you seen coming to america ? or the naked gun hot shots ! |

| | |
|---|---|
| SEEKER: | yes , those were pretty good |
| HUMAN: | how about spaceballs , i thought this was pretty funny . |
| HRED: | have you seen foxcatcher ? it 's a comedy about a man on a true story |
| OURS: | have you seen dumb and dumber to ? it 's a classic ! |

| | |
|---|---|
| SEEKER: | haha that one was great |
| HUMAN: | okay , let ' me see if i can offer one more . |
| HRED: | have you seen foxcatcher ? it 's a comedy about a man that turns out on a true story . |
| OURS: | i have n't seen that one yet . i 'll have to look it up . |

| | |
|---|---|
| SEEKER: | awesome |
| HUMAN: | robin hood : men in <unk> ? this one had me rolling . |
| HRED: | have you seen foxcatcher ? it 's about a man who has a rich guy who is a <unk> rich guy |
| OURS: | have you seen monty python 's life of brian ? that 's a good one ! |

| | |
|---|---|
| SEEKER: | haha i forgot about that one ! i 'll be watching that one tonight |
| HUMAN: | great . |
| HRED: | it stars nanjiani , zoe kazan , holly hunter , ray romano , adeel akhtar , and anupam <unk> no , i have n't seen it yet . |
| OURS: | have you seen caddyshack ii ? it 's a good one . |

| | |
|---|---|
| SEEKER: | thanks for all of your help i hope you have a great day |
| HUMAN: | welcome , glad to help . have a nice day . good bye |
| HRED: | you 're welcome ! have a great day ! |
| OURS: | you 're welcome ! have a great day ! |

Table 5: Sample conversation from validation set

| | |
|---|---|
| SEEKER: | hi ! |
| HUMAN: | hi ! how is your morning going ? |
| HRED: | hi , what kind of movies do you like |
| OURS: | hello ! what kind of movie are you looking for ? |
| SEEKER: | great , thanks ! i was thinking of starting the day off with a great comedy , any suggestions ? |
| HUMAN: | i love the hangover it always makes me laugh . and i bet lots of people are hungover this morning ! airplane ! is another classic of mine . |
| HRED: | that 's a good one ! have you seen the lego movie ? it 's a great movie ! |
| OURS: | i have n't seen that one ! what kind of movies do you like ? i can help you with that . |
| SEEKER: | i 've seen the hangover it 's really good ! i thought about watching knocked up have you seen it ? |
| HUMAN: | i enjoyed that movie . i enjoyed spy it 's a well done parody . |
| HRED: | it 's about a man who gets revenge on a cyborg and a hockey team . it 's really good . it 's really good . i also like the girl on the train |
| OURS: | i have n't seen that one . i will have to check it out . i hope i was able to help you find a good one ! have a great day ! |
| SEEKER: | thanks ! i 'll check it out ! |
| HUMAN: | sure ! have a great day ! |
| HRED: | you 're welcome ! have a great day ! |
| OURS: | have you seen revenge of the nerds ? it 's really good . it 's a classic . |
| SEEKER: | you too . |

Figure 5: 2D embedding of movies in our conversation database. The edge weight in the similarity matrix is proportional to the number of co-occurrences in the same dialogue. Left: all movies, colored by number of occurrences from light blue (low) to red (high). Right: names of movies with highest number of occurrences. Embedding via Jacomy et al. [28].

Table 6: Sample conversation from validation set

| | |
|---|---|
| HUMAN: | good day ! |
| HRED: | hi what kind of movies do you like |
| OURS: | hi , how can i help you ? |
| SEEKER: | hey i 'm really into romance movies any good suggestions ? thanks ! |
| HUMAN: | hmm . let me think for a sec . have you seen the big sick ? it 's romantic and funny |
| HRED: | what kind of movies do you like to watch ? have you seen the conjuring 2 ? |
| OURS: | what kind of movies do you like ? i 'm good , and you ? |
| SEEKER: | i like movies like beauty and the beast and a walk to remember oh and lol i have never seen the the big sick or never been kissed and romeo and juliet |
| HUMAN: | i highly recommend ! great story ! have you seen the notebook ? that 's a big one <unk> romance fans |
| HRED: | i have n't seen that one yet but i 've heard good things about it ! |
| OURS: | i have n't seen that one yet but i have heard good things about the longest ride and the fault in our stars |
| SEEKER: | i love the the notebook a really great film . |
| HUMAN: | and if you liked romeo and juliet then i have to reccommend titanic for leo . also a recent sweet romantic comedy was the duff |
| HRED: | i have n't seen that one , but i 've heard good things about it ! |
| OURS: | i have n't seen that one yet but i 've heard good things about it have you seen message in a bottle ? |
| SEEKER: | ahh the great titanic what a classic thanks for recommending that one . i will definitely have to check out the the duff thanks so much for your help ! |
| HUMAN: | no problem ! i have one more to suggest . phantom thread ! it 's new and i did n't like it as much but a lot of others do ! it <unk> romance in a unique way no problem at all ! i hope you enjoy the films you decide to watch and enjoy the rest of your day ! |
| HRED: | you 're welcome ! have a great day ! |
| OURS: | you 're welcome ! have a great night ! |
| SEEKER: | thanks i will defiantly check out the phantom thread i really appreciate it and you have a great day ! |

Table 7: Sample conversation from validation set

| | |
|---|---|
| SEEKER: | hello |
| HUMAN: | hello |
| HRED: | hi , how can i help you ? |
| OURS: | hello , what kind of movie are you looking for ? |
| SEEKER: | how are you this evening ? |
| HUMAN: | what type of movies do you like ? good , thanks |
| HRED: | what kind of movies do you like ? i 'm good |
| OURS: | can i help you find a movie ? |
| SEEKER: | i am looking for a musical . what do you suggest ? |
| HUMAN: | have you seen enchanted ? maybe you 'd like la la land . |
| HRED: | have you seen get out yet ? it 's pretty good |
| OURS: | i have n't seen that one yet but have heard good things about it have you seen prometheus ? |
| SEEKER: | no i have n't seen that one . i have been meaning to watch it though . oh that 's a good suggestion . what about something older ? |
| HUMAN: | the rocky horror picture show is a great one . |
| HRED: | i have n't seen that one yet but heard good things about ender 's game i have n't seen it yet but i have n't seen it yet but i 've heard of it . |
| OURS: | i 'm trying to think of something along the same way . sweeney todd : the demon barber of fleet street is another good one |
| SEEKER: | i 've seen that before when i was younger . i did n't know whether to laugh or be scared . lol |
| HUMAN: | fiddler on the roof maybe ? sorry , i have n't seen many musicals |
| HRED: | i have n't seen that one . i will have to check it out . thanks for chatting ! |
| OURS: | i have n't seen grease but i have heard good things about it . |
| SEEKER: | yes ! that 's great ! thank you for the suggestions ! i try to watch them soon , especially enchanted ! hope you have a good night ! |
| HUMAN: | thanks you too |
| HRED: | you 're welcome ! have a great night ! |
| OURS: | no problem ! have a good day ! |