[Reviews · NeurIPS 2018]

Reviewer 1



Summary: This paper provides a large-scale data set of 10,000 conversations for the domain of movie recommendation. Strengths: + The authors will release the first large-scale dataset for conversational movie recommendations, which can be significant for follow-up work on conversational recommendation. + The authors propose an end-to-end trainable architecture, with sub-components of new neural-based models for sentiment analysis and cold-start recommendations, along with natural language. Weaknesses: - As this work has the perspective of task-oriented recommendation, it seems that works such as [] Li, Xiujun, et al. "End-to-end task-completion neural dialogue systems." arXiv preprint arXiv:1703.01008 (2017). are important to include, and compare to, at least conceptually. Also, discussion in general on how their work differs from other chatbox research works e.g. [] He, Ji, et al. "Deep reinforcement learning with a natural language action space." arXiv preprint arXiv:1511.04636(2015). would be very useful. - It is important that the authors highlight the strengths as well as the weaknesses of their released dataset: e.g. what are scenarios under which such a dataset would not work well? are 10,000 conversations enough for proper training? Similarly, a discussion on their approaches, in terms of things to further improve would be useful for the research community to extend -- e.g. a discussion on how the domain of movie recommendation can differ from other tasks, or a discussion on the exploration-exploitation trade-off. Particularly, it seems that this paper envisions conversational recommendations as a goal oriented chat dialogue. However, conversational recommendations could be more ambiguous.. - Although it is great that the authors have included these different modules capturing recommendations, sentiment analysis and natural language, more clear motivation on why each component is needed would help the reader. For example, the cold-start setting, and the sampling aspect of it, is not really explained. The specific choices for the models for each module are not explained in detail (why were they chosen? Why is a sentiment analysis model even needed -- can't we translate the like/dislike as ratings for the recommender?) - Evaluation -- since one of the contributions argued in the paper is "deep conversational recommenders", evaluation-wise, a quantitative analysis is needed, apart from user study results provided (currently the other quantitative results evaluate the different sub-modules independently). Also, the authors should make clearer the setup of how exactly the dataset is used to train/evaluate on the Amazon Turk conversations -- is beam-search used as in other neural language models? Overall, although I think that this paper is a nice contribution in the domain of movie conversational recommendation, I believe that the authors should better position their paper, highlighting also the weaknesses/ things to improve in their work, relating it to work on neural dialogue systems, and expanding on the motivation and details of their sub-modules and overall architecture. Some discussion also on how quantitative evaluation of the overall dialogue quality should happen would be very useful. == I've read the authors' rebuttal. It would be great if the authors add some of their comments from the rebuttal in the revised paper regarding the size of the dataset, comparison with goal-oriented chatbots and potential for quantitative evaluation.

Reviewer 2



The paper describes a model for building a chat bot where the bot is to recommend movies to users via natural language dialog. The model mixes dialog modeling, sentiment characterization, and movie recommendation in a modular architecture that connects an HRED (for dialog modeling with context) to a hierarchical encoder (for sentiment analysis) as well as an auto-encoder based recommender module. The recommender module can be pre-trained by an external dataset. Section 3 describes a process to create a collection of 10K conversations focused on recommending movies using Amazon Mechanical Turk. The process is by itself interesting and could serve as a good example for similar data collection needs -- after the workers generate the natural language data, they are also asked to provide symbolic annotation for sentiment and whether the recommendation seeker has seen the movie. Such input serves as ground truth for the relevant inference tasks. The paper addresses a broad, new class of dialogs that is practically important and technically challenging, and is thus worthy of attention in this conference. The model is well motivated and is shown to perform well compared to simpler or isolated models for the individual tasks. The paper is well written and describes many detailed considerations in the model set-up and the experimentation process. Some discussions on how such a model could be adapted to address other tasks would be interesting. As the paper sets out to explore conversational recommendation systems as a class of problems, it will be useful to summarize the experience learned in this particular task (and dataset), and highlight which parts are expected to be applicable to a wider domain, and the conditions on such domain that are pre-requisites for the method to work.

Reviewer 3



This paper is about a conversational (movie) recommendation system. The model is a neural network encoder-decoder style for next response generation, but with additional components which identify the 'recommendation seeker's' sentiment and select the movie to talk about accordingly. The paper also releases a dataset of 10,000 recommendation conversations in the domain. The previously good models for dialog generation are based on hierarchical encoder-decoder frameworks and is the base models used in this work. The ideas here involve tracking sentiment towards suggestions to pick out next suggestions and for generating the text. I found the model interesting and the work is a reasonable approach to the problem. The focus on tracking entities and entity sentiment on the movies is very likely to improve generation performance. The weakness of the paper is that the evaluation is a bit disappointing and lacking in discussion. The paper presents confusion matrices for sentiment prediction, a human study for quality of next utterance, and a rmse for movie recommendations (the last one is a bit unclear). I assume the last is about whether the right movies were spoken about. The use of ratings then is not clear, the paper must clarify what is being predicted and the evaluation metric. Anyway, the results do not confirm a definite improvement over previous approaches, both the human evaluation and recommendations. The paper does not provide any discussion of the results obtained and what could be improved. Morever, the generation capabilities are not tested, the fluency of the utterances etc. In addition, the conversation as a whole must be evaluated possibly with humans, the paper only has a utterance level comparison. Moreover, it is unusual to use real dialogues by replacing recommender utterances, the context of the dialog so far, could have resulted in different paths for the model leading to equally good but different conversations. Why not evaluate with users inputing to the conversation or sample the seeker utterances but evaluate the complete dialog? The main problem with the paper is inconclusive evaluation. After author response: I thank the reviewers for their response. It would be important to update the paper based on questions/comments of all reviewers and also add more discussion. It is an interesting paper overall.